# Altered Tryptophan-Kynurenine Pathway in Delirium: A Review of the Current Literature

**DOI:** 10.3390/ijms24065580

**Published:** 2023-03-15

**Authors:** Ang Hui Phing, Suzana Makpol, Muhammad Luqman Nasaruddin, Wan Asyraf Wan Zaidi, Nurul Saadah Ahmad, Hashim Embong

**Affiliations:** 1Department of Emergency Medicine, Faculty of Medicine, Universiti Kebangsaan Malaysia, Jalan Yaacob Latif, Bandar Tun Razak, Cheras, Kuala Lumpur 56000, Malaysia; 2Department of Biochemistry, Faculty of Medicine, Universiti Kebangsaan Malaysia, Jalan Yaacob Latif, Bandar Tun Razak, Cheras, Kuala Lumpur 56000, Malaysia; suzanamakpol@ppukm.ukm.edu.my (S.M.);; 3Department of Medicine, Faculty of Medicine, Universiti Kebangsaan Malaysia, Jalan Yaacob Latif, Bandar Tun Razak, Cheras, Kuala Lumpur 56000, Malaysia

**Keywords:** acute brain dysfunction, delirium, inflammation, kynurenine pathway, neuroinflammation

## Abstract

Delirium, a common form of acute brain dysfunction, is associated with increased morbidity and mortality, especially in older patients. The underlying pathophysiology of delirium is not clearly understood, but acute systemic inflammation is known to drive delirium in cases of acute illnesses, such as sepsis, trauma, and surgery. Based on psychomotor presentations, delirium has three main subtypes, such as hypoactive, hyperactive, and mixed subtype. There are similarities in the initial presentation of delirium with depression and dementia, especially in the hypoactive subtype. Hence, patients with hypoactive delirium are frequently misdiagnosed. The altered kynurenine pathway (KP) is a promising molecular pathway implicated in the pathogenesis of delirium. The KP is highly regulated in the immune system and influences neurological functions. The activation of indoleamine 2,3-dioxygenase, and specific KP neuroactive metabolites, such as quinolinic acid and kynurenic acid, could play a role in the event of delirium. Here, we collectively describe the roles of the KP and speculate on its relevance in delirium.

## 1. Introduction

Delirium is a form of acute brain dysfunction that represents a change from the patient’s baseline cognitive and behavioral functioning. It is characterized by disturbances in attention, awareness, and multiple aspects of cognition, posing a significant burden across all healthcare settings, including in emergency departments (EDs) and intensive care units (ICUs). The current diagnosis of delirium is based on a diagnostic and statistical manual-5 (DSM-5) which has stated five key criteria for delirium diagnosis, (1) a disturbance in attention and awareness, (2) the disturbance develops over a short period from hours to days and represents a change from the patient’s baseline attention and awareness, (3) with additional disturbance in cognition, (4) the disturbances are not explained by a pre-existing or other neurocognitive disorder, and (5) the disturbance is a direct physiological consequence of a wide variety of insults [1]. Based on the clinical psychomotor presentation, delirium is divided into three subtypes, such as hypoactive (unarousable, lethargy), hyperactive (agitated) and mixed subtype [2]. Evidence suggests that delirium is linked to several adverse short- and long-term outcomes, including a 2.2-fold increase in in-hospital mortality [3], a 2.5 time increase in health care costs [4], and a 12-fold increase in the risk of dementia [5,6,7]. 

Delirium is a multifactorial condition, as it depends on the interaction between predisposing and precipitating factors. The condition occurs more commonly in older people, although it can occur in all age groups [8]. Evidence reported that approximately 15% to 30% of geriatric patients presented to the ED with delirium [9] and 27.6% of older patients developed delirium during admission to medical wards [10]. Delirium in older adults is contributed by one or more predisposing factors, such as cognitive impairment, hearing impairment, nursing home residence, and a history of stroke [11]. These risk factors suggest the phenomenon of a vulnerable brain, defined as a normal physiologic brain with reduced protection and/or adaptive response to insults [12]. Meanwhile, the precipitating factors for delirium vary depending on the clinical setting. Infection, followed by drugs and electrolyte imbalance are common precipitating factors for delirium in community-dwelling elderly individuals admitted to acute care units [13]. In the ICU setting, delirium is caused by the use of mechanical ventilation, hypoxia, infection, electrolyte imbalance, sedative administration and surgical procedures [14]. Post-operative delirium (POD) has been recognized as one of the most common post-surgical complications, particularly in older adults. The incidence of POD depends on the type of surgery and increases with the severity of the surgical insults [15]. In a meta-analysis of delirium post hip fracture, Yang and colleagues (2017) reported an accumulated incidence of 24% [16], whereas in post-operative cardiac bypass, the incidence is up from 26% to 52% [17]. The condition is triggered by many precipitating factors that can be divided into pre-operative (such as age, baseline cognitive function, and underlying comorbidities), intraoperative (such as type of surgery and urinary catheter insertion) and post-operative factors (such as pain, and electrolyte imbalances) [18,19]. These numerous factors may all trigger delirium whereby the effect of any risk factor on POD is complex and not fully elucidated. 

The diagnosis of delirium relies on a thorough clinical evaluation that often goes undetected, and the current management mainly treats the underlying etiology. Several clinical assessment tools have been validated to improve the recognition of delirium in clinical settings, such as the confusion assessment method (CAM), CAM-ICU, and 4-AT [20]. However, POD and delirium in the ED are frequently misdiagnosed as depression or dementia because of similarities in their active presentations, especially in the hypoactive subtype [21,22]. Delirium could be a manifestation of severe illnesses, such as sepsis, and missed diagnosis of delirium will lead to several adverse outcomes, including delay in initiating medical therapy and inappropriate disposition [23]. 

The pathophysiology of delirium remains poorly understood, and several hypotheses have been postulated, including neuroinflammation, neuronal aging, oxidative stress, neurotransmitter deficiency, and network dysconnectivity [24]. The neuroinflammatory hypothesis of delirium remains the most widely explored, whereby there is compelling evidence that suggests acute systemic inflammation from peripheral stimulus exerts delirium via the release of several pro- and anti-inflammatory cytokines which may affect brain function. Previous reports suggest that inflammation occurring in the periphery causes systemic inflammation which may trigger neuroinflammation [25,26]. Neuroinflammation is widely recognized as chronic, as opposed to systemic inflammation, which can be manifested in acute and chronic forms. Table 1 summarizes the differences between systemic inflammation and neuroinflammation. Chronic inflammation in the brain is associated with several neurodegenerative conditions, such as Alzheimer’s disease and Parkinson’s disease. Acute systemic inflammation may exacerbate underlying neuroinflammation and make the brain more vulnerable to acute working memory and behavior deficit, which has important implications for delirium superimposed on dementia. Although, reports support the neuroinflammatory role in delirium, the mechanism by which it affects brain activity is poorly described. Voils and colleagues (2020) suggest that increased inflammation, the upregulation of the indoleamine dioxygenase (IDO) enzyme, and accumulation of neurotoxic metabolites from the tryptophan—kynurenine (TRP-KYN) pathway contribute to the pathogenesis of delirium [27]. The increase in inflammatory cytokines significantly shifts the TRP metabolism to the kynurenine pathway (KP) metabolites production that carries several biological functions which can lead to neuronal cell injury, excitotoxicity, and apoptosis [28]. 

Knowing that the underlying mechanism of delirium is multifactorial, an understanding of the universal pathway leading to delirium is crucial to allow for more specific and sensitive biomarkers that can demonstrate the clinical utility of delirium intervention. The KP could be a promising metabolic pathway implicated in the pathogenesis of delirium. In addition, there remains a need to further understand the systemic role of the altered KP in response to acute insults (infection, trauma) and possible neurotoxicity and neurological disruption associated with the KP neuroactive metabolites. For that reason, this review was carried out to gain a better understanding of the role of the TRP-KYN pathway in systemic inflammation-induced delirium, and its significance in elucidating the pathogenesis and intervention for delirium. 

## 2. Tryptophan-Kynurenine Pathway 

The KP is a metabolic pathway of the essential amino acid TRP degradation, and the pathway generates several neuroactive metabolites. Tryptophan is mainly obtained from protein-based foods and plays an important role in brain homeostasis as the precursor to serotonin (5-HT) neurotransmitters and melatonin hormone. In human and animal cells, the majority of TRP (95–99%) is converted to KYN and its metabolites through the KP, and only a small fraction of TRP is degraded through the 5-HT pathway [31]. The KP is upregulated in response to immune system activation, and produces various neuroactive metabolites that are able to influence the function on brain innate immune cells [32]. The first phase of TRP degradation via the KP is catalyzed by two main enzymes, tryptophan 2,3 dioxygenase (TDO) and indoleamine 2,3-dioxygenase (IDO) [33,34] (Figure 1). IDOs are primarily distributed in extrahepatic tissues, such as the brain, blood, kidney, and lung and the highest levels of TDO are expressed in the liver [35,36]. TDO is responsible for the regulation of systemic TRP concentrations, and its expression is activated by corticosteroids and glucagon. Meanwhile, IDO is stimulated by pro-inflammatory cytokines, such as interferon-gamma (IFN-γ), interleukin-1Beta, -6, (IL-1β, IL-6), tumor necrosis factor-alpha (TNF-α), and reactive oxygen species (ROS) during immune stimulation [37]. With aging, the activity of TDO in the brain decreases while the activity of IDO increases [36,38]. 

Kynurenine is then catabolized to three routes, to 3-hydroxykynurenine (3-HK) by kynurenine-3-monooxygenase (KMO), to kynurenine acid (KYNA) by kynurenine-amino-transferase (KAT), and to anthranilic acid (AA) by kynureninase [39,40]. The 3-HK is catabolized to hydroxyanthranilic acid (HAA) and progresses in two different routes; the oxidation pathway in which ATP is formed in the liver and the degradation pathway in which quinolinic acid (QA) and picolinic acid (PA) are formed. Quinolinic and picolinic acid are neurotoxic compounds and act as N-methyl-D-aspartate (NMDA) receptor agonists and oxidative radicals while KYNA is claimed to be neuroprotective by being an NMDA receptor antagonist [41]. Under acute inflammatory situations, such as trauma, surgery, and sepsis, the KP is highly upregulated and dysregulation in the balance of neuroprotective and neurotoxic metabolites of KYN possibly leads to neuronal damage and delirium.

## 3. Role of the Tryptophan-Kynurenine Pathway in Delirium

Acute peripheral inflammation is known to trigger delirium [42]. The KP forms an intriguing link between peripheral inflammation and brain function capable of precipitating neuroinflammatory conditions and promoting cognitive decline and behavior disturbances. The release of pro-inflammatory cytokines triggers the activation of the KP that leads to the accumulation of metabolites with immune suppression, neuroactive, and pro-oxidative activities, potentially having a direct effect on neuron and microglial activities. Moreover, KYN and its metabolites can influence neurotransmitter systems. 

Previous studies have shown that peripheral inflammatory stimuli induce a profound immunological response in the brain via microglia activation. KYN is the first metabolite produced from the primary precursor L-TRP and an increase in the KYN/TRP ratio reflects the activity of IDO. Acute stress and inflammation from the periphery increase plasma and brain KYN [43,44], and over 60% of KYN in the brain is delivered from the peripheral circulation [45]. KYN from the plasma is easily transported to the brain by the large neutral amino acid transporter 1 (LAT-1) and organic anion transporter 1 and 3 [46,47]. In the brain, KYN is metabolized to KYNA in astrocytes or to 3-HK, AA and QA in microglia [48]. Microglia regulate KP balance, and their activation during the acute phase of inflammation can accelerate the conversion of TRP into neurotoxic and pro-oxidative metabolites, such as QA [49]. Although modest, outcomes from a clinical study demonstrated that autopsy results for patients who died from delirium had a significantly higher number of CD-68 in the brain, which indicates the presence of microglial activation [50,51,52]. Meanwhile, animal studies reported an increase in the expression of microglial Iba1 and astrocytic glial fibrillary acidic protein (GFAP) following systemic lipopolysaccharide (LPS) administration [53,54,55].

Modest clinical studies have implicated kynurenine pathway activation in the pathogenesis of delirium (Table 2), although there is no causal proven between these relationships. The diagnosis of delirium was made based on clinical assessment and is guided by standard criteria, such as CAM and DSM-V. Wilson and colleagues (2012) previously demonstrated a significant association between increasing plasma KYN levels and the KYN/TRP ratio with the incidence of delirium and death in a medical ICU setting [56]. In this study, 84 critically ill patients were recruited, and 36 (43%) had sepsis upon admission. Following adjusting for age, sedative agents and severity of illness, higher levels of KYN and KYN/TRP ratio were both independently associated with fewer days free from delirium and coma (*n* = 71). Several clinical studies emphasized a strong relationship between sepsis severity with increased IDO activity. Delirium is a common cerebral manifestation in sepsis and is associated with increased morbidity and mortality. A study by Darcy and colleagues (2011) indicated that plasma KYN significantly increased in sepsis and predicted ICU mortality based on a quantitative scoring index, such as the sequential organ failure assessment (SOFA) and acute physiology and chronic health evaluation (APACHE II) score [57]. These findings suggested that the KP compounds, such as KYN and neuroactive metabolites, could be useful biomarkers to predict delirium in critically ill patients with sepsis. In the meantime, Nettis and colleagues (2020) observed sickness behavior and mood changes after being injected with subcutaneous IFN-α and these differences in behavior were associated with increased levels of KYN/TRP ratio [58]. Acute sickness behavior is an early manifestation during infection and the symptoms include depressed mood, anhedonia, and at times, delirium. Meanwhile, in a cohort of elderly patients in geriatric wards (*n* = 81), Egberts and colleagues (2016) investigated the association between aspirin use and TRP levels in delirium and non-delirium patients [59]. Aspirin downregulates TRP degradation via the regulation of cytokine signaling and could protect against systemic inflammation-induced delirium. The study identified that the levels of TRP were lowered in delirium patients and the use of aspirin was neither associated with the levels of TRP nor delirium.

Post-operative and critically ill patients in ICU often have elevated levels of pro-inflammatory cytokines and these mediators upregulate the KP to produce neuroactive metabolites, such as QA and KYNA. In surgical ICU patients, several studies have reported associations between the incidence of delirium after trauma and POD with KP metabolites. In a more recent study, Voils and colleagues (2020) reported that the plasma KYNA levels were significantly elevated in a cohort of older adults with delirium after major trauma [27]. In addition, they also observed a significant decrease in TRP levels and a higher KYN/TRP ratio in patients with delirium. Meanwhile, in a large multicenter study, Watne and colleagues (2022) reported a significant increase in the concentration of QA in the cerebrospinal fluid (CSF) of 224 hip fracture patients with delirium [61]. Among all subjects (*n* = 450), the study identified that 22% of patients died at the end of one year after surgery, and this was strongly associated with QA levels. The study is the first to analyze perioperative CSF of KP metabolites in a cohort of surgical patients. Previously, Jonghe and colleagues (2012) observed that the plasma KYN/TRP ratio that was taken perioperatively was significantly elevated in delirium of elderly hip fracture patients [60]. Although the study did not observe a significant association between TRP or KYN levels with delirium, they concluded that the KYN/TRP ratio could allow for early detection of patients at risk to develop delirium. 

The current hypothesis regarding the mechanism of POD is mainly from animal studies, with limited evidence from clinical studies, including neurotransmitter deficiency, neuroinflammation and prior cerebral vascular events [63]. Previous investigations have revealed lower levels of TRP in elderly patients with POD [64,65]. These studies imply that the low TRP levels are associated with POD via a decrease in serotonin (5-HT) synthesis in the brain. However, post-operative supplementation with TRP did not prevent the incident of POD in older adults undergoing major elective surgery [66]. It might be possible that increased TRP degradation during surgery is due to activated IDO activity which converts TRP to KYN. In a targeted metabolomics analysis of plasma patients at day 2 after surgery, Tripp and colleagues (2021) detected multiple metabolites linked to POD, including a significant increase in KYN and KYNA, and a decrease in TRP levels [62]. 

The current understanding of mechanisms of delirium involving the KP based on preclinical research is largely unexplored, contributed by lack of established clinically relevant animal models of delirium. However, many animal studies documented the role played by the immune system and the KP on cognition and behavior. Several animal models have been developed based on acute peripheral inflammatory exposure, including administration of single or repeated endotoxins [54,67,68,69,70,71,72,73,74,75,76,77,78,79,80,81,82,83,84], cecal ligation and puncture (CLP)-induced sepsis-associated encephalopathy [53,85,86], bile duct ligation (BDL) model of hepatic encephalopathy [87], administration of immunostimulant polyinosinic: polycytidylic (Poly I:C) acid [88,89], CD40 agonist antibody [90], and living organisms, such as Bacillus Calmette–Guerin (BCG) [91]. These studies were chosen based on recommendations that animal models of delirium should be precipitated by etiological factors that are known to cause delirium and exhibit acute cognitive and behavioral changes seen in human delirium [92]. Ideally, the model demonstrates a fluctuating course [93]. Table 3 summarizes the effects of acute systemic inflammation on KP activity and behavior from in-vivo studies. All studies documented an increase in the KP activity via the measurement of the IDO enzymes, the KYN/TRP ratio, and several neuroactive KP metabolites, including QA and KYNA. 

Several test batteries for the evaluation of acute cognitive and behavior disturbances were utilized, using either a single or a combination test to assess for sensory-motor, emotional, and/or cognitive function. These studies documented acute changes in the locomotor and exploratory activities [53,67,68,71,74,75,76,77,79,80,81,82,85,86,87,88,89,90,91], symptoms of anhedonia and depression [67,71,72,73,74,75,77,78,79,80,81,84,87,88,90,91], acute decline in cognitive function [53,54,68,82,83,86,87] and anxiety-like behavior [53,83,84,87,88], which are key symptoms of delirium. Regardless of the severity of the peripheral insults, acute systemic inflammation significantly affects central KP activity and generates acute behavioral and cognitive disturbances. Several studies showed that even mild acute systemic inflammation can induce sickness behavior that is associated with peripheral and central KP activation and its downstream metabolites [68,69,74,77,94], and the symptoms are shown to be exaggerated in aged animals [91]. 

The activation of the central KP by peripheral LPS challenge resulted in drastic changes in behavior and it is dose-dependent. In the porcine model of sickness behavior, a single peripheral LPS administration was associated with decreased TRP and increased KYN levels in the plasma but not in the brain [95]. In contrast, Larsson and colleagues (2016) demonstrated that systemic inflammation following a repeated LPS administration induces more robust changes in the peripheral and central KP activity than single LPS administration [96]. In the context of sepsis-associated encephalopathy (SAE), which is a cerebral manifestation commonly occurring in sepsis and one of many causes of delirium depicting robust changes in the KYN levels. Sepsis induces activation of cerebral endothelial cells over the blood-brain barrier and releases various inflammatory mediators into the brain. Evidence suggests that SAE is aggravated by neurotoxicity and neuroinflammation, which may contribute to neuronal death. Activation of IDO by IFN-γ, TNF-α, IL-6, and IL-1β in the hippocampus generates several neuroactive compounds, including QA and 3-HK that can modulate cognitive and behavior changes, affecting the development of delirium during infection.

### 3.1. Role of Indoleamine 2,3-Deoxygenase in Delirium

Under acute inflammatory conditions, increased levels of pro-inflammatory cytokines and oxidative stress may stimulate IDO, subsequently activating the catabolism of TRP via KP. Animal and human studies have shown that the plasma KYN/TRP ratio is significantly increased, which indicates that IDO and TRP catabolism are upregulated in delirium. The finding is supported by several in-vivo studies that documented a significant increase in the expression of brain IDO [54,69,71,75,76,80,81,84,87]. 

IDO plays an important role in immunoregulation and is implicated in suppressing the effector phase of the immune response. Under normal physiological conditions, IDO provides natural defense against inflammation because of its ability to restrict T-cell function and promotes immune tolerance. Under pathological conditions, IDO becomes activated and exerts anti-inflammatory activity via direct T cell suppression from degradation of TRP or by enhancement of local Treg-mediated immunosuppression. Interestingly, accumulating evidence implicates IDO in the immune escape of cancer cells, autoimmune and chronic inflammation, such as rheumatoid arthritis and human immunodeficiency virus (HIV) infection [97,98]. Moreover, recent evidence highlights the role of IDO in severe COVID-19 and sepsis [99]. In delirium patients, direct measurement of IDO activity is yet to be performed. 

Several studies have demonstrated the effects of IDO1-inhibitor in the modulation of the central KP activity and attenuated acute inflammation-induced behavior and cognition disturbances. Kynurenine is the first metabolite of TRP degradation by IDO enzymes and pre-treatment with an IDO-inhibitor increases the plasma and/or brain levels of TRP and lowers the KYN/TRP ratio. Jiang and colleagues (2018) demonstrated that IDO-inhibitor prevented BDL-induced learning and memory impairment in rats [87]. Similarly, Comim and colleagues (2017) demonstrated that an IDO inhibitor (1-methyltryptophan) blocked the enzymatic activity triggered by sepsis in the hippocampus and preserved the habituation memory [85]. Further, Gao and colleagues (2016) showed that treatment with an IDO-inhibitor attenuated cognitive impairment in animals with sepsis induced by CLP [53]. Severe sepsis and septic shock are regarded as a cytokine-mediated hyper-inflammatory phase, usually associated with enormous immune dysregulation. Previous in-vivo studies indicated that pharmacological inhibitors of IDO were shown to upregulate chemokine expression and reduce mortality from CLP-induced sepsis [100]. Chemokines may act directly on T cells or via indirect effects on antigen-presenting cells (APCs), which subsequently promote differentiation of T cells to the site of inflammation. Contrary to the porcine model of shock, Wirthgen and colleagues (2018) indicated that treatment with 1-methyltryptophan (1-MT) in the early phase of immune response did not significantly downregulate IDO activity, instead they observed increased levels of TRP and KYNA levels in the plasma, peripheral organ, and brain tissues [101]. The author suggested that the lack of IDO inhibition of 1-MT and the potential weakness of antimicrobial activity should be considered in future applications, especially in sepsis. 

Meanwhile, IDO induction by the nuclear-binding domain (NOD)-1 and NOD-2 agonist have been shown to enhance plasma kynurenine levels and sickness behavior associated with LPS administration [79]. From these studies, the KYN/TRP ratio seems to be a relevant clinical biomarker for both prognostic and therapeutic purposes in systemic inflammation-induced delirium. However, in most animal studies of acute behavioral and cognitive disturbances, a brain tissue sample was frequently obtained rather than the plasma and imposed limitations on the availability of non-invasive samples in furthering the understanding of disease mechanisms. 

### 3.2. Role of Neuroactive KP Metabolites in Delirium

The KP metabolites are associated with delirium via both microglial activation and subsequent neuronal injury. Microglia plays an important role in the regulation of brain development, and maintenance of the neuronal network and function. Increased IDO immunoreactivity in microglia will lead to the production of both neuroprotective and neurotoxic metabolites that could induce neuronal injury and subsequently affect neuronal function. Of all of the metabolites of the KP, the QA and KYNA have been the most studied that are associated with cognitive and behavior consequences in acute systemic inflammation. Both QA and KYNA act on N-methyl-d-aspartate (NMDA) receptors. 

#### 3.2.1. Quinolinic Acid

Under normal physiological conditions, the production of QA from the main precursor TRP mainly involves the synthesis of NAD^+^. Under the pathological situation, QA exerts neurotoxicity activity via multiple mechanisms, including direct neurotoxicity by acting as an agonist at the NMDA receptors, production of ROS, acting as a potent lipid peroxidant, and inhibits glutamate uptake by astrocytes [102,103]. Quinolinic acid acts selectively at NMDA receptors, specifically on the NR2A and NR2B subunits, and elevates cytosolic calcium depositions, thus exerting necrosis and apoptosis of neurons [104]. A previous in-vitro study has shown that QA application in hippocampus tissue was associated with massive deposition of Ca^+^ in damaged mitochondria [105]. Calcium (Ca^+^) is a universal second messenger and participates in neural transmission when a neuron is activated. Overstimulation of NMDA receptors increase intracellular Ca^+^ and leads to the activation of Ca^+^-dependent lysis enzymes which results in neuronal death [106]. This information is supported by recently published data that demonstrated the causal association of QA with neuronal damage marker, neurofilament light chain protein (NfL) in the cerebrospinal fluid (CSF) of patients with delirium [61]. 

Concomitantly, QA showed pro-oxidative stress activity due to its ability to generate ROS formation. Recently, a study by Hosoi and colleagues (2021) demonstrated that intrastriatal administration of QA in rats induces acute microglia oxidative damage and mitochondrial dysfunction which indicates an acute neurotoxicity effect of QA on the brain via oxidative stress [107]. The study obtained a functional image of acute oxidative damage in the brain that occurred 3 to 24 h after QA injection. In contrast, QA does not readily cross the BBB and systemic administration of this metabolite observed only small concentrations in the brain, which subsequently converted to NAD^+^ [108]. In the brain, QA induces oxidative stress on its ability to stimulate NMDA receptors or interacts with free iron ions via the Fenton reaction to generate oxidizing agents, including ROS [109]. Iron is a vital element for all photosynthetic organisms and serves as a cofactor to enzymes in oxidation. Additionally, acute stress, such as during sepsis, downregulates the nuclear translocation of antioxidant defense factors, such as nuclear factor (erythroid-derived 2)-like 2 (Nrf2), further predisposing neuronal injury and apoptosis [110,111]. Nrf2 is recognized as a key regulator of antioxidants in the CNS and serves as a neuroprotection in response to oxidative stress [112,113]. Nrf2 declines with age and is further exposed to oxidative stress during acute inflammation [114]. This phenomenon could explain the vulnerable brain theory in the occurrence of delirium in the elderly.

Oxidative stress may cause additional neurotoxicity effects of QA via lipid peroxidation and inhibit the re-uptake of glutamate by astrocytes. Animal studies report that intrastriatal injection of QA into rats was associated with lipid peroxidation within 2 h after exposure, and the finding was supported by the presence of higher concentrations of hydroxyl radical in the striatum [115,116]. Lipid peroxidation is one of the characteristic features of systemic inflammation associated with KP activation, producing highly reactive aldehydes, such as acrolein, malondialdehyde (MDA), and hydoxynonenal (HNE) [117]. QA increases the formation of lipid peroxidation, mediated by an NMDA receptor or by its interaction with ferrous ions to form QA-Fe^2+^ complexes [118,119]. There is evidence that lipid peroxidation occurs in response to systemic inflammation, including sepsis and surgery. Toufekoula and colleagues (2013) demonstrated that MDA levels in septic patients were correlated with the presence of organ dysfunctions [120]. Meanwhile, one study reported a greater concentration of plasma MDA that was taken immediately after cardiopulmonary bypass surgery and the levels were positively correlated with a marker for cerebral injury, S100β [121]. Meanwhile, changes in neurotransmitter systems are thought to contribute to the occurrence of delirium and the KP is directly related to several neurotransmitter systems, particularly glutamatergic and serotonergic systems [122]. The essential amino acid TRP is the precursor of 5-HT, which plays a vital role in the CNS, particularly in mood regulation. Although, only a small fraction of TRP is metabolized via the serotonin pathway, small changes in the production of 5-HT do alter brain morphology, physiology, and behavior. In animal studies, activation of IDO by inflammatory cytokines alters serotoninergic and glutamatergic neurotransmission. Acute and chronic administration of LPS significantly changed the concentration of TRP via KP activation, increasing the KYN/TRP ratio and decreasing the 5-HT/TRP ratio [72]. The activation of the KP by pro-inflammatory cytokines potentially contributes to a substantially increased amount of glutamate via the release of QA in activated microglia and reduces glutamate reuptake by astrocytes. Increased amounts of glutamate released can continuously activate several ligand-gated ion channels receptors, such as NMDA receptors, α-amino-3-hydroxy-5-methylisoxazole-4-propionate (AMPA), and kainic acid (KA) receptors, which eventually leads to an increase in intracellular calcium loads in neurons [123]. All of this evidence suggests that glutamate activation contributes to neuronal dysfunction and death, supporting QA-induced neurotoxicity during systemic inflammation. 

Although evidence has supported the behavioral effects of QA, the use of an NMDA receptor antagonist in eliminating symptoms associated with delirium is anecdotal. Animal studies suggest that ketamine administration was capable to attenuate LPS-induced depressive-like behavior and these behavioral effects were associated with a decrease in QA in the brain [74,124]. However, Walker and colleagues (2013) demonstrated that ketamine administration failed to abrogate sickness behavior associated with LPS administration in rats, despite a reduction in the KYN levels [74]. Meanwhile, in human studies, pre-operative administration with ketamine has been shown to provide neuroprotective effects on limiting the development of delirium, contributed by its anti-inflammatory actions [125].

#### 3.2.2. Kynurenic Acid

KYNA possesses neuroprotective activity from its ability to suppress the immune response and acts as a non-competitive antagonist at the NMDA, AMPA, and KA receptors. With regards to the immunomodulatory roles of KYNA, Ferreira and colleagues (2020) observed that intrastriatal injection of KYNA in rats ameliorated the increase in ROS, pro-inflammatory cytokines, and glutathione peroxidase activity induced by QA [126]. Meanwhile, systemic administration of KYNA in rats prevents the accumulation in the neutrophil extracellular traps (NETs), a marker for BBB permeability and protects the brain from mitochondrial dysfunction during sepsis [127]. Previous studies demonstrated that KYNA facilitates immunosuppression, by activating the aryl hydrocarbon receptor (AhR) [128,129]. AcR signaling has multiple impacts on the immune response and functions on ligand-activated transcriptional factors. Evidence reported that KYNA is one of the endogenous ligands for AhR and works synergistically in modulating inflammatory cytokines and oxidative stress [130]. Moreover, the antagonistic action of KYNA on NMDA, AMPA, and KA receptors provides additional neuroprotective benefits via the prevention of glutamate neurotoxicity induced by QA [131]. Inhibition of KMO, which is the main enzyme in the catabolism of KYN to neurotoxic metabolites, possibly creates a therapeutic opportunity in preventing delirium during systemic inflammation. Several animal studies demonstrated that peripheral KMO inhibitor administration elevates levels of KYNA in the brain and improves cognition associated with neuronal damage, indicating that the intervention shifts the TRP degradation into a neuroprotective pathway [132]. 

On the contrary, an attempt to elucidate the impact of exogenous KYNA on cognition and behavior observed that exogenous administration of L-kynurenine (100 mg/kg) in rats led to spatial memory deficits [133]. Changes in brain KYNA levels are assumed to affect neurotransmitter release via modulation of the non-competitive blockade of a7-nicotinic acetylcholine (a7nACh) receptors [134]. However, findings from several experimental studies opposed the assumption, whereby KYNA had no effect on the a7nACh receptors [135,136,137]. To date, the relationship between KYNA and cognitive deficit remains unclear. The author suggested that KYNA-induced spatial memory deficits could occur through the blockade of NMDA or a7nAch receptors in the prefrontal cortex. Both NMDA and a7nAch receptors are mainly identified in the hippocampus, and antagonist-mediated blocking of NMDA receptors by KYNA leads to excessive excitatory neurotransmitters, such as glutamate and Ach in other brain regions [138]. Curiously, the administration of kynurenine aminotransferase (KAT II) inhibitors has been shown to prevent the elevation of KYNA levels in the brain and block cognitive dysfunction [139]. In human studies, KYNA was found to be upregulated in the blood of patients with ICU delirium and POD [27,62]. In this context, increased serum KYNA may be attributable to an excessive secretion of pro-inflammatory cytokines, which subsequently aggravates the KP activity as seen in critically ill patients in ICU settings. 

## 4. Translational Implications

The current management of delirium requires proper detection, recognition of the cause, and treatment of symptoms. To date, there are no proven biomarkers as diagnostic or prognostic indicators for delirium. There are many potential biomarkers that have been investigated for their potential utility in delirium, including inflammatory biomarkers, neuroimaging, and markers of neuronal injury. In fact, some of these biomarkers are proven to be associated with the incidence of delirium (Table 4). However, the elevated levels of inflammatory mediators may not be specific for delirium and their concentrations rise in other conditions associated with inflammation. Considering the limitation, further research is needed to investigate the predictive utility of these biomarkers.

Kynurenine pathway and its biological active metabolites are potentially useful as a biomarker in the assessment of delirium risk, diagnosis, and prognosis. In delirium, several kynurenine parameters have been measured, including TRP, KYN, KYN/TRP ratio, QA, and KYNA. Previous evidence suggests that older patients experiencing POD seem to have an elevated perioperative KYN and reduced TRP levels [27,64,65]. Aging is associated with increased TRP degradation towards the KP pathway and is implicated in several age-related disorders. Westbrook and colleagues (2020) reported the KYN/TRP ratio was positively correlated with functional decline and frailty status [141]. Measurement of KYN and TRP, together with the KYN/TRP ratio can help to predict the degree of delirium risk, especially in older adults admitted for surgery, or critically ill patients in the ED and ICU. However, the clinical utility of TRP levels alone in the prediction of POD is arguable, considering contradictory findings obtained from a sample of hip fracture patients [60]. 

The diagnosis of delirium is made clinically and requires basic investigative work-up that is directed from history and physical findings, including an infective marker, blood and urine, and measurement of electrolyte levels. Diagnosis is often difficult in patients presenting for the first time, and their baseline cognitive function is not readily available. Previously, Watne and colleagues (2022) reported that patients with delirium had considerably higher levels of CSF-QA compared with those without delirium [61]. In addition, an acute intrastriatal injection of QA in the brain elicited several motor and cognitive deficits associated with NMDA receptor-mediated neurotoxicity, oxidative stress, and tau phosphorylation [103]. These mechanisms are involved in the development of delirium. As the QA concentration rises in acute and chronic neurodegenerative conditions, the best approach for use of QA in delirium diagnosis is perhaps through the use of age-stratified cut-off points. In the meantime, critically ill patients with delirium and POD are associated with poor clinical outcomes and greater resource consumption [3,4,5,6,7]. Delirium patients who died during hospitalization were found to have higher KYN and KYN/TRP compared to the survivors [56]. These results are in line with current findings that indicate a positive correlation between KYN and the KYN/TRP ratio with poor neurological outcome and mortality after cardiac arrest [142], and suggest their potential to predict the clinical outcome in delirium. Until now, these biomarkers have yet to be proven in their utility in clinical practice.

Furthermore, there are promising results in preclinical models that suggest the KP as a therapeutic target for delirium, perhaps evaluated as monotherapy or in combination with other immunotherapeutic agents. Given that the activation of the KP has been demonstrated in delirium, inhibition of IDO might have the potential to reduce inflammation in delirium associated with acute illness, such as sepsis. Recently, several IDO inhibitors have been evaluated, but these have been limited in patients with advanced malignancies [143]. Meanwhile, inhibition of KMO has been shown to increase brain production of KYNA and improve cognition in animal models [144]. KYNA is neuroprotective, and the level increases with KMO inhibition or enhancement of KAT enzymes. However, overexpression of KYNA could be detrimental to cognition and some studies have reported increased levels of KYNA in the plasma sample of delirium patients. Therefore, furthering preclinical and clinical research is required to develop a better understanding of this relationship and may open a therapeutic opportunity for delirium interventions.

## 5. Future Research Perspectives

Future research in delirium should address several issues. Further research into this pathway should provide a clearer system-wide view of the general mechanisms underlying both chronic neurodegenerative disorders and transient phenomena, such as delirium, ideally improving the understanding of the pathophysiologic links between the acute and chronic neurological conditions. Specifically, the potential role of IDO1 in the development of newer therapeutic strategies targeting neuroinflammation is worth exploring given the burden of cognitive deficit disorders in an increasing ageing population. In addition, improvements in animal models representing delirium should facilitate elucidating the exact pathophysiological and biochemical mechanisms leading to the disease state, and clinical presentations. 

Meanwhile, the recent advances in more precise diagnostic tools in the case of delirium provides opportunities for novel precision medicine techniques in the detection, treatment, and optimization of patients presenting acutely with delirium and early cognitive disorders. Serial measurements of these delirium biomarkers can possibly reveal the true natural history of disease progression, as opposed to only relying on markers of established injury to neuronal tissue. Delirium can be understood as brain failure in the setting of systemic dysfunction due to multiple causes that may not necessarily originate from the nervous system. Therefore, it may not be absolutely necessary to obtain direct samples, such as CSF, to investigate the condition. The availability of plasma delirium biomarkers avoids more invasive diagnostic measures that can equally contribute to early diagnosis and improvement of therapeutic interventions. 

## 6. Conclusions

In conclusion, our review supports the speculation that the KP is implicated with acute cognitive and behavior disturbances associated with systemic inflammation in delirium. The KP has a potential role in preclinical research and clinical utility in pathogenesis discovery and future interventions for acute delirium. There are, however, considerable knowledge gaps in relation to the role and functions of individual KP neuroactive metabolites in the disease process.

## Figures and Tables

**Figure 1 ijms-24-05580-f001:**
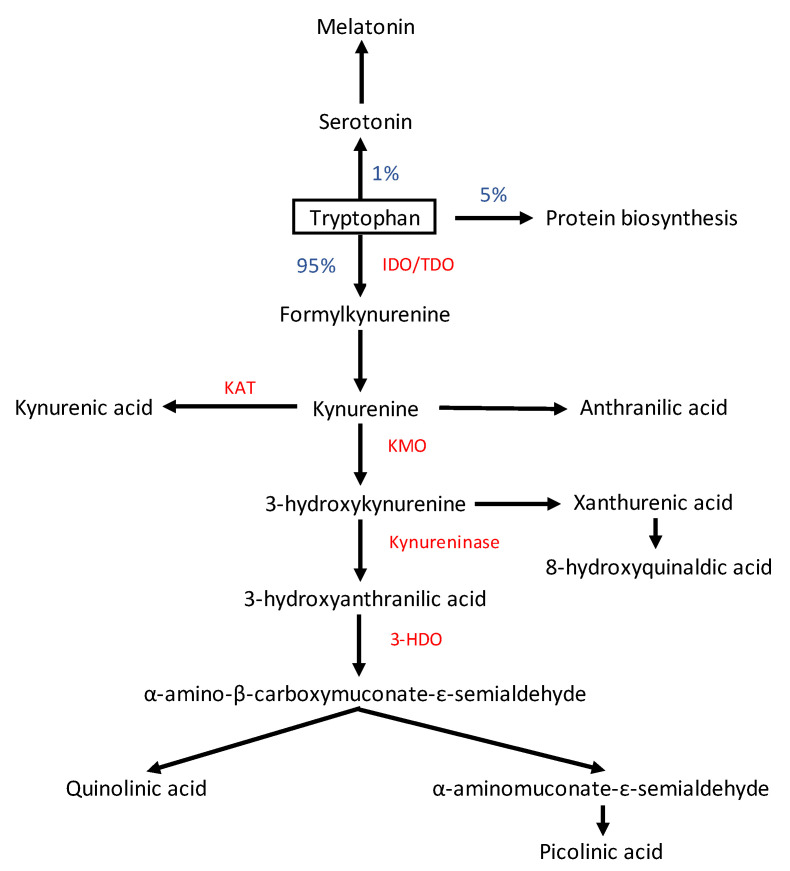
The kynurenine pathway of tryptophan (TRP) metabolism. Note: enzymes are highlighted in red. IDO: indoleamine 2,3-dioxygenase; TDO2: tryptophan 2,3-dioxygenase; KAT: kynurenine aminotransferases; KMO: kynurenine 3-monooxygenase; 3HDO: 3-hydroxyanthranilic acid oxygenase.

**Table 1 ijms-24-05580-t001:** Differences between systemic inflammation and neuroinflammation [29,30].

	Systemic Inflammation	Neuroinflammation
Location	Throughout the body	Within the central nervous system (CNS) and spinal cord
Onset	Acute and chronic	Widely regarded as chronic
Innate immune cells	Macrophages, neutrophils, dendritic cells	Microglia and astrocytes
Causes	Trauma, sepsis, surgery	Autoimmune disease, metabolic disorders, aging, peripheral sources
Symptoms	Pain, increased temperature, redness, swelling	Memory deficit, depression, frailty
Associated conditions	Acute inflammation, such as trauma and infection Chronic inflammation, such as systemic lupus erythematosus and multiple sclerosis	Alzheimer’s disease and Parkinson’s disease

**Table 2 ijms-24-05580-t002:** Altered kynurenine pathway in humans with delirium from recent clinical studies.

Author/Year	Study Design	Type of Sample/Metabolites Measured	Sample Size/Population Details	Delirium Assessment Tool	Age-Matched (Y/N)	Findings
Wilson et al. (2012) [56]	Observational	Plasma (TRP, KYN)	84 (71 with brain dysfunction, 13 no brain dysfunction) in ICU	CAM	Y	Higher KYN levels and KYN/TRP ratio in patients with brain dysfunction (both coma and delirium) than in normal group; Increased KYN/TRP levels were associated with fewer days alive and free from delirium and coma than in normal group.
Voils et al. (2020) [27]	Case-control	Plasma (TRP and KYN from the targeted metabolomics analysis)	130 (65 patients with delirium, 65 control) in ICU	CAM-ICU	Y	Significant increase in KYNA in delirium group; TRP levels decreased in delirium group; KYN/TRP ratio was significantly higher in delirium group; Quantitative enrichment analysis from the global metabolomic analysis indicated 3-fold enrichment in the TRP metabolic pathway in the delirium group.
Jonghe et al. (2012) [60]	Prospective cohort	Plasma (TRP, KYN)	140 (71 with delirium, 69 without delirium) hip fracture patients	CAM	Y	Delirium group was older (mean age 85 years old, *p* < 0.05); No significant differences in the levels of TRP and KYN during the course of delirious and non-delirious state; Significantly higher perioperative KYN/TRP ratio in delirium group.
Watne et al. (2022) [61]	Observational	Plasma and CSF (TRP, and KYN metabolites from the targeted metabolomics analysis)	450 (224 with delirium, 226 without delirium) hip fracture patients	DSM-V criteria	Y	Delirium group had a tendency to have prefracture cognitive impairment; QA levels in CSF is significantly higher in delirium group; No significant difference in CSF concentrations of glutamate and aspartate in delirium and non-delirium group S; CSF QA is significantly associated with higher NfL.
Egberts et al. (2016) [59]	Cross-sectional	Plasma (TRP)	80 (22 had delirium, 58 without delirium) elderly in internal medicine and geriatric wards.	DSM-IV criteria	Y	Delirium patients were older than non delirium patients (mean age 85 years old vs. 80 years old, *p* < 0.05); Plasma TRP levels were lower in delirious than non-delirious group (statistical analysis was not reported); No significant different in plasma neopterin and TRP levels in delirium patients who used aspirin than delirium patients who did not use aspirin.
Tripp et al. (2021) [62]	Case-control	Plasma (TRP and KYN metabolites from the targeted metabolomics analysis)	104 (52 with delirium, 52 control) older adults undergoing major non-cardiac surgery	CAM	Y	Significant increase in KYN and KYNA, and reduced TRP levels in delirium cases at post-operative day 2, as compared to non-delirium group.

Abbreviations: Y: yes; N: no; ICU: intensive care unit; CAM: confusion assessment method; CAM-ICU: confusion assessment method-intensive care unit; DSM: diagnostic statistical manual of mental disorders; KYN: kynurenine; TRP: tryptophan; KYNA: kynurenic acid; QA: quinolinic acid; CSF: cerebrospinal fluid; Nfl: neurofilament light polypeptide.

**Table 3 ijms-24-05580-t003:** The effect of acute peripheral inflammation on the kynurenine pathway activation and behavior from in vivo studies.

Author	Type of Exposure	Type of Subjects	Test Battery	Follow up Duration	Findings
Comim et al., 2017 [85]	Sepsis induced by cecal ligation and perforation (CLP)	Wistar rat (60 days old)	Open field test (OFT)	24 h and day 10	IDO inhibitor prevented the changes in the mitochondrial respiratory chain enzymatic activity in the hippocampus caused by sepsis; Habituation memory was preserved in the sepsis group receiving IDO inhibitor.
Jiang et al., 2018 [87]	Laparotomy + bile duct ligation (BDL)	Male Wistar rats (220–240 g)	Sucrose preference test, forced swimming, marble burying test, and elevated plus maze	0, 7, 14, 21, and 28 days	Mice exhibit anxiety behavior, a decline in learning and memory function, and locomotor started from day 7 after surgery and symptoms were attenuated by IDO inhibitor; Elevated expression levels of mRNA pro-inflammatory cytokines (TNF-α, IL-1β, and IL-6) in the brain 14 days post BDL and day 7 in the serum; Constant elevation of IDO-1 and IDO-2 expression in the hippocampus and cerebral cortex, especially 14 days after BDL; The ratio of KYN/TRP increased and 5-HT/TRP decreased in the hippocampus and cerebral cortex in the BDL surgery group. These changes were reversed by 1DO inhibitor treatment; A high level of quinolinic acid was observed on day 28 post-BDL.
Cathomas et al., 2015 [90]	Single i.p. CD40 agonist antibody (CD40AB) injection	Male C57BL/6J mice (10–14-week-old)	Saccharine consumption, fear conditioning, and locomotor activity	Day 2, 4, 5, 6, 7, 8, and 12 after CD40AB injection	CD40AB induces behavioral effects; decreased saccharin preference, consumption, and operant responding; without affecting locomotor activity or unconditioned stimulus fear and treadmill running; The behavioral effects were led by increased TNF-α levels in the serum and IFN-γ levels in the serum and brain; Increased pro-inflammatory cytokines were followed by increased kynurenine metabolites (KYN, 3-HK, and QUIN) on days 1–7/8 in the serum and brain; Co-injection with TNF-α blocker, Etanercept blocked CD40AB effects on behavioral and the kynurenine pathway; Oral administration of a selective IDO inhibitor prevented the activation of the kynurenine pathway but did not influence acute or extended sickness behavior.
Gomes et al., 2018 [67]	Single i.p. LPS 0.33 mg/kg	Male C57BJ/6J mice (24-month-old)	Open field test, and sucrose preference test	24 h	Decreased the locomotor activity in the LPS group than in the control; Increase in pro-inflammatory cytokines (IL-1β, TNF-α, and IFN-γ) induced by LPS in the hippocampus, striatum, and prefrontal cortex; Decrease in the brain levels of 5-HT and 5-HIAA induced by LPS; LPS significantly decreases KYNA levels and KYNA/KYN ratio (neuroprotective branch) in brain areas; LPS induced an increase of 3-HK and QA levels (neurotoxic branch).
Agostini et al., 2020 [68]	Single intravenous LPS 100 μg/kg	Male and female APPswe/PS1dE9 mice model of Alzheimer’s disease and WT (4.5-month-old)	Food burrowing, and Y-maze test	24 h, 48 h, and 96 h	LPS suppressed food burrowing activity in WT males and females and APP/PS1 females; LPS suppressed exploration of Y-maze in all experimental groups, without altering spatial working memory performance; LPS led to a significant increase in plasma levels of IL-6 in all groups, while TNF-α was significantly elevated only in LPS-treated WT females. Plasma levels of IFN-γ were unaltered. Anti-inflammatory IL-10 was significantly higher in LPS-treated female WT and APP/PS1; Increased hippocampal TRP and end metabolites of the 5-HT and KYN pathways 4 h after LPS injection; Immunohistochemistry revealed that LPS had no effects on microglial density in any hippocampal areas (measured by the percentage area covered by Iba1 positive microglial and the number of microglial cells per mm^2^).
Gomes et al., 2020 [69]	Single i.p. LPS 1mg/kg	Male Wistar rats (3-month-old)	Locomotor activity, rotarod test, and forced swimming test	24 h	Body weight loss and the decrease in locomotor activity induced by LPS; CUR-LNC modulated the levels of pro-inflammatory and anti-inflammatory cytokines (IL-1β, TNF-α, IL-6, and IL-10) in the hippocampus; Up-regulated in IDO-1 and IDO-2 mRNA expression in the hippocampus.
Tufvesson-Alm et al., 2020 [70]	Two doses i.p LPS 0.83 mg/kg 16 h apart	FVB/N and C57BL/6J mice (3–4-month-old)	Open field test, fear conditioning, and Y-maze	24 and 48 h	Mice treated with repeated administration of LPS showed reduced locomotor activity and increased anxiety-like activity; Repeated LPS in FVB/N mice caused an increase in brain KYNA levels as compared to saline-treated mice; Repeated administration of LPS however, did not cause a deficit in the working memory.
Kang et al., 2011 [71]	Single i.p. LPS 0.8 mg/kg	CD-1 mice 10–12-week-old	Open field test, sucrose preference test, and forced swimming test	24 h	Body weight and food intake significantly reduced after the LPS injection; No significant difference in locomotor activity from the OFT after the LPS injection; Mice injected with LPS exhibited reduced sucrose preference tests; Hippocampal mRNA expression proinflammatory cytokines (IL-1β, IL-6, and TNF-α) increased 24 h post-LPS administration.
Heisler & O’Connor 2015 [54]	i.p. LPS 0.5 mg/kg	C57BL/6J mice 12–16-week-old (group into wild type, IDO^−/−^, KMO transgenic mice)	Novel object recognition (NOR)	24 to 48 h	LPS challenge induced deficit in novel object recognition (NOR) in WT mice; IDO^−/−^ mice challenged with LPS were secured from NOR deficit; The glial activity was upregulated as indicated by increased mRNA expression of microglial Iba1 and astrocytic GFAP 24 h post LPS challenge; Increased the expression of the pro-inflammatory cytokines IL-1β and TNF-α and reduction in IL-6 in both WT and IDO^−/−^; Increased mRNA expression of IDO1 48 h post-LPS challenge for WT, a contrast to IDO^−/−^ mice; Acute administration of peripheral kynurenine attenuated depressive-like behaviors; KMO^−/−^ mice administered with LPS were secured from a deficit in NOR.
Guo et al., 2016 [72]	Single i.p. LPS 3mg/kg Vs repeated i.p LPS 500 μg/kg every other day for 2 weeks	Sprague Dawley rats, 230–280 g	Sucrose preference test and forced swimming test	2, 4, 6,8, and 10 h	Repeated LPS administration was associated with a significant reduction in body weight, decreased sucrose preference test and locomotor activity; A rapid increase in glutamate release in the hippocampus in both acute and repeated LPS challenges; Acute LPS administration raised the hippocampal concentrations of TRP, 5-HT, 5-HIAA, and KYN; Repeated LPS challenge did not affect hippocampal TRP concentration but increased the concentration of KYN and the KYN/TRP ratio; The acute LPS challenge reduced the KYNA concentrations, decreased the 5-HT/TRP ratio, and increased the KYN/TRP ratio.
Murray et al., 2015 [89]	i.p. Poly I:C (12 mg/kg) ± IL-6 (50 μg/kg)	Female C57BL6/J mice (WT And IFNAR1^−/−^)	Open field test and food burrowing	24 h to 96 h	Poly I:C induced sickness behavior in WT mice (decrease in the number of rears in the open field, weight reduction); Higher plasma kynurenine level and KYN/TRP ratio in WT than IFNAR1^-/-^ mice 24-h post poly I:C challenge; Reduced expression of IDO in IFNAR1^−/−^ mice; KP metabolites were undetected in the hippocampus or frontal cortex 3 h post poly I:C challenge.
Gao et al., 2016 [53]	Cecal ligation and perforation (CLP) induced animal model of sepsis	Male C57BL/6 mice	Open field test and fear conditioning test	12 h and day 14	CLP significantly increased the levels of KYN and KYN/TRP ratio and expression of microglial Iba1 in the hippocampus; CLP increased the expression of pro-inflammatory cytokines, TNF-α, IL-1β, and IL-6 in the hippocampus; Treatment with IDO inhibitor, 1-MT downregulated the increased expression of KYN, KYN/TRP ratio, and IDO activity and downregulated marker of microglial activation induced by CLP; Treatment with IDO inhibitors (1-MT and L-TRP) attenuated CLP-induced cognitive impairment; Treatment with L-KYN induced cognitive deficit in sham mice but did not affect the locomotor activity and anxiety-like activity.
Golia et al., 2019 [73]	i.p LPS 0.33 and 0.83 mg/kg	Male C57BL/6 mice 12–15-week-old	Not specified	3 h	Decreased food intake both in high and low-dose LPS; LPS increased hippocampal expression of inflammatory cytokines (IL-1β), TNF-α, IL-6, and COX-2; Acute treatment with a high and low dose of LPS impaired neural plasticity- increased expression of inflammatory markers; IL-1β, PGE2; The modulation of TDO2 was not affected by LPS.
Gibney et al., 2013 [88]	i.p. Poly I:C 6mg/kg	Male Sprague Dawley 250–350 g	Home cage activity test, saccharin preference, and open field test	6, 2-, 48, and 72 h	Significantly reduced body weight at 24 h post poly I:C administration; Reduced locomotor activity and anxiety-like behavior 6 h post poly I:C administration; Increased level of IL-1β, IL-6, TNF-α, and CD11b at 6 h time point in frontal cortex and hippocampus; A significant 70-fold increase in IDO expression in the frontal cortex and a 3.3-fold change in the hippocampus at 6 h post poly I:C administration; Increased in KYN/TRP ratio in the brain at 24, 48 and 72 h post poly I:C administration; KYN level in the hippocampus was significantly higher at 24 and 48 h post poly I:C administration, and back to control levels by 72 h; The concentration of TRP increased in the frontal cortex at 24 and 48 h and 6, 24, and 48 h in the hippocampus after poly I:C injection; No alteration in 5-HT concentration in the frontal cortex and hippocampus.
Kelley et al., 2013 [91]	i.p. Bacillus Calmette–Guerin (BCG) i.p. 10^8^ CFU/mice	Adult mice Balb/c 4–6-month-old vs. aged mice 20–24-month-old	Locomotor activity, sucrose preference test, and tail suspension test	1, 7, 14 or 21 days	Aged mice exhibited a persistent reduction in body weight (BW) as compared to adult mice with a transient reduction in BW; Aged mice exhibited sickness behavior (reduced locomotor activity) until day 7 and recovered 2 weeks after infection; Locomotor activity was not affected in adult mice after BCG infection; Longer rearing and anhedonia in aged mice than in adult mice after BCG infection; Aged mice demonstrated an increase in KYN/TRP ratio as compared to the saline group and the levels were significant at 7,14, and 21 days after BCG infection.
Walker et al., 2013 [74]	LPS i.p 1mg/kg (dose to induced depressive-like behavior) LPS i.p. 0.83 mg/kg (dose to induce acute sickness response)	CD-1 (6-week-old) C57BL/6J (12-week-old)	Locomotor activity, sucrose preference test, and forced swimming test	2–28 h	Significant increase in the brain KYN/TRP ratio in LPS-treated mice compared with control; LPS also significantly induced the elevation of KP metabolites, such as QA; LPS decreased body weight, food consumption, and motor activity (sickness behavior), and the effect was not altered by ketamine; LPS decreased sucrose preference (depressive behavior), which was blocked by ketamine pre-treatment; Increase in the brain IL-6 and IL-1β at 6 and IL-6 at 28-H p.i. and the effects were not altered by ketamine; LPS increased plasma KYN/TRP ratio at 6 and 28 h, and brain KYN/TRP ratio at 28 h after treatment, and the effects were not affected by ketamine treatment.
Zhao et al., 2019 [75]	i.p LPS 0.5 mg/kg and/or unpredictable chronic mild stress (UCMS)	Adult male C57BL/6J mice, 8–12-week-old	Open field test, forced swimming test, and tail suspension test.	24 h	No significant differences in locomotor activity in both LPS and UCMS mice; LPS induced more apparent depressive-like behaviors than UCMS; LPS induced robust expression of TNF-α, IL-1β, and IL-6 in the serum and brain areas (prefrontal cortex, hippocampus, and striatum), as compared to UCMS mice; IDO expression in the brain (prefrontal cortex and hippocampus) was significantly increased following LPS and UCMS; The decrease of 5-HT and BDNF was detected only in the hippocampus of LPS-stressed mice.
Schneiders et al., 2015 [76]	i.p LPS 50 or 2500 μg/kg	C/EBPβ^+/+^ (Wild type) and C/EBPβ^-/-^ (KO) mice 6–12 weeks old	Locomotor activity (two-compartment cage)	8 or 24 h	LPS stimulation showed drop in locomotor activity in response to normal environmental stress (NES) after 2 h, while the activity of KO mice remained significantly high; The plasma level of IL-6 and IL-10 of both WT and KO increased 8 h p.i. with high dose LPS and returned to basal levels at 24 h in WT mice. (TNF-α was not detected); Higher expression of inflammatory mediators (IL-6, TNF-α, IL-10 in the brains of KO mice) 24 h after LPS stimulation as compared to WT; LPS induced significant upregulation of IDO-mRNA expression in the hypothalamus 8 h p.i in WT which was absent in KO mice; Tryptophan-hydroxylase (TPH) 2 was significantly increased after LPS treatment in KO mice, but not in WT; mRNA expression of IDO and TPH2 in the hypothalamus returned to basal levels at 24 h in both genotypes; The mRNA expression of KMO was unchanged at 8 h and significantly higher at 24-h p.i. in KO mice compared to WT mice; The IDO expression in the liver was significantly higher in KO mice 8 h p.i. compared to WT mice and no difference in both genotypes at 24 h.
Dinel et al., 2014 [77]	i.p LPS 5ug	Male db/db and db/+ mice between 10 and 12 weeks	Two-compartment cage and forced swimming test	2 h to 25 h	LPS significantly increased the duration of immobility 24 h post-LPS in db/+; LPS significantly increased the brain KYN/TRP ratio and this increase was significantly reduced in db/db mice; LPS significantly increased plasma levels of IL-1b, TNF-a, and IL-10 in db/+ and db/db mice 2 h after treatment; LPS-treated db/db mice exhibited similar peripheral levels of KYN to their db/+ counterparts but lower brain KYN levels; LPS significantly increased hippocampus mRNA expression of IL-1β, TNF-α, IL-6, IFN-γ, and IL-10 in both db/+ and db/db mice.
Lu et al., 2021 [78]	i.p LPS 1.0 mg/kg	Male C57BL/6J mice, 4–5 weeks of age	Sucrose preference test, forced swimming test, and tail suspension test	24 h	LPS-induced depression-like behavior; Mice administered with LPS demonstrated the elevation of the TNF-α, IL-6, IL-1β, IL-10, TRP, and KYN levels and lower 5-HT in the hippocampus.
Farzi et al., 2015 [79]	i.p. LPS (0.1 or 0.83 mg/kg)	Male C57BL/6N mice, aged at 10 weeks	Sucrose preference test, open field test, forced swimming test, and tail suspension test	21 h	Aggravation of sickness behavior induced by FK565 (NOD1 agonist) or MDP (NOD2 agonist) in combination with LPS; The sickness behavior was paralleled by enhanced plasma and cerebral mRNA levels of proinflammatory cytokines (IFN-c, IL-1b, IL-6, TNF-a), as well as enhanced plasma levels of kynurenine.
Zhang et al., 2018 [80]	i.p. LPS (0.5 mg/kg) and/or unpredictable chronic mild stress (UCMS)	Cat C overexpression (CAT C OE) and Cat C knockdown (CAT C KD) transgenic mice together with wild type (WT) at 8-weeks-old	Open field test, forced swimming test, and tail suspension test	24 h	Marked increase in IDO levels in the hippocampus and prefrontal cortex of LPS and UCMS mice; Cat C OE induced peripheral and central inflammatory response (promoted microglia/macrophage activation) with significantly increased TNFα, IL-1β and IL-6 in serum, hippocampus, and prefrontal cortex; Cat C OE promoted upregulation of IDO and downregulation of 5HT expression in the brain, and subsequently aggravated depression-like behaviors.
Carabelli et al., 2020 [81]	i.p. LPS 250μg/kg	Male Wistar rats, 60 days-old	Open field test and forced swimming test	24 h	LPS led to a marked weight loss; LPS induced a depressive-like behavior, and the symptoms were blocked by 1-methyltryptophan (IDO inhibitor); LPS increased 5HIAA/5HT and IDO and decreased 5-HT levels in the hippocampus.
Peyton et al., 2019 [82]	i.p. 2X LPS (0.25 mg/kg, 0.50mg/kg, or saline) 16 h apart	Male C57Bl/6J mice, three months old	Pavlovian conditioning, accelerated rotarod radial 8-arm maze, and prepulse inhibition	24 h, 48 h, and 144 h	Mice treated with peripheral low-dose LPS (0.25 mg/kg) exhibited significant increases in pre-cortex KYN and KYNA; Dual LPS administration did not impair exploratory activity; Dual LPS-treated mice showed deficits in reference memory while working memory was observed to be normal.
Danielski et al., 2018 [86]	Cecal ligation and perforation (CLP)	Male Wistar rats, approximately 60 days old, 250–350 g	Open field test and object recognition test	24 h-day 10	Pro-inflammatory cytokine (TNF-α, IL-1β, and IL-6) levels in the hippocampus significantly increased 24 h after sepsis induction; KYN in the cortex, but not in the hippocampus significantly increased 24 h after sepsis induction.
Imbeault et al., 2020 [83]	Single i.p LPS 0.83mg/kg	Male C57Bl6/NCrl mice aged 13 to 18 weeks	Elevated plus maze and fear conditioning test	24–96 h	LPS-induced anxiety-like behavior and cognitive deficits; Changes in behavior were accompanied by an increase in brain and plasma KYN: TRP ratio; Pre-treatment with IDO1 inhibitor (1-MT) 7 days before LPS injection reduced the levels of KYN and the KYN: TRP ratio in the brain; Pre-treatment with TDO2 inhibitor (680C91) did not change the levels of KYN and KYN: TRP ratio in the brain and serum, and KYNA levels in the serum; Pre-treatment with IDO1 inhibitor and TDO2 inhibitor failed to reduce anxiety parameters and mitigate cognitive deficit produced by LPS.
Choubey et al., 2019 [84]	Single i.p. LPS 0.83 mg/kg and/or restraint stress (RS)	Adult male Swiss albino mice	Open field test and forced swimming test	3–24 h	Acute systemic inflammatory exposure to LPS, RS, and RS plus LPS induced anxiety-like behavior and depression-like behavior; LPS, RS, and RS plus LPS augmented the activation of the TLR-4 receptor that induces the release of pro-inflammatory IL-1β in the hippocampus region; Exposure to RS, LPS, and RS plus LPS significantly upregulated NF-ƙB and IDO-1 gene expression in the hippocampus region.

Abbreviations: IDO: indoleamine2,3-deoxygenase; TDO: tryptophan 2,3-dioxygenase; KMO: kynurenine 3-monooxygenase; TPH: tryptophan-hydroxylase; BDL: bile duct ligation; TNF-α: tumor necrosis factor-alpha; IL-1β: interleukin-1beta; IL-6: interleukin-6; IL-10: interleukin-10; IFN-γ: interferon-gamma; KYN: kynurenine; TRP: tryptophan; 5-HT: 5-hydroxytryptamine (serotonin); 5-HIAA: 5-Hydroxyindoleacetic; KYNA: kynurenic acid; 3-HK: 3-hydroxykynurenine; QA: quinolinic acid; CD40AB: CD40-antibody; LPS: lipopolysaccharide; WT: wild type; CUR-LNC: curcumin loaded nanocapsule; OFT: open filed test; NOR: novel object recognition; GFAP: glial fibrillary acidic protein; BDNF: brain-derived neurotrophic factor; Poly I:C: polyinosinic:polycytidylic acid; CLP: cecal ligation/perforation; 1-MT: 1-methyltryptophan; COX-2: cyclooxygenase-2; BCG: Bacillus Calmette–Guerin; CFU: colony forming unit; BW: body weight; UCMS: unpredictable chronic mild stress; KO: knockout: NES: normal environmental stress; NOD: nucleotide-binding oligomerization domain; MDP: muramyldipeptide.

**Table 4 ijms-24-05580-t004:** Summary of biomarkers associated with delirium [140].

Biomarker	Sample Type	Main Effects
CRP	Plasma	CRP levels was found to be associated with incidences of delirium (high CRP levels) and recovery from delirium (low CRP levels); Higher plasma CRP levels were associated with risk, duration, and severity of delirium.
S-100B	Plasma	Patients with delirium showed higher S-100B levels in the serum; Higher S-100B levels were associated with higher delirium incidence and severity.
NSE	Plasma	Higher NSE levels were associated with postoperative delirium; Elevated NSE levels were significantly associated with risk of delirium and mortality.
Systemic inflammatory marker	Plasma	IL-6, -8, -10, and TNF-α levels were positively associated with higher delirium incidences and severity.
Neuroimaging	MRI/EEG	MRI- white matter hyperintensities; EEG- generalized theta and delta slowing correlated with delirium severity.

Abbreviations: CRP: C-reactive protein; S-100B: S100 calcium-binding protein B; NSE: neuron specific enolase; IL: interleukin, TNF-α: tumor necrosis factor-alpha; MRI: magnetic resonance imaging; EEG: electroencephalography.

## Data Availability

Not Applicable.

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
