# Peer review of "Altered Tryptophan-Kynurenine Pathway in Delirium: A Review of the Current Literature"

_ijms, 2023, doi:10.3390/ijms24065580_

Round 1
Reviewer 1 Report
This article summarizes the role of KP, and speculates on the correlation between KP in the pathogenesis of delirium, and summarizes the limitations of existing research. This article has certain reference value and useful guideline . And there are still some problems in the work of this article that need to be explained or corrected.
1. The table 1 occupies a large space, but the content is less, and there is no good comparison of the literature provided. It is recommended to refer to relevant literature for modification , such as Neuroscience and Biobehavioral Reviews 123 ( 2021 ) 1-13.
2. There is weak contact between content of Table 1 and text in this article. The references in the table are basically intended to appear in this table and are not reviewed in the text.
3. The structure and content of the article are casual. There are a lot of contents in the first-level directory, the second-level directory and the third-level directory, and there is no obvious primary and secondary distinction. It is recommended to refer to J Neuroinflammation 18,135 ( 2021 ) for modification.
4. In the conclusion section. 'One of the main limitations of our review is related to challenges associated with the identification of potentially relevant literature. The failure of an animal-study-based model for pure delirium further underscores the limitations of this review. ' This should be a summary of the limitations of current research, not the limitations of this review.
Reviewer 2 Report
Phing et al. presented altered tryptophan-kynurenine pathway in delirium. This is an interesting review article, suggesting that the kynurenine pathway is implicated in acute cognitive and behavior disturbances during systemic inflammation relevant to delirium.
The paper is well written. There are, however, a few issues to be addressed to further improve the article.
1. The paper does not fully argue the relationship between IDO activity and immune systems. More detail discussion of IDO and regulatory T cells and effector T cells will be intriguing.
2. As to the altered tryptophan-kynurenine pathway in human with delirium, the number of clinical studies cited in Table1. is too small.
Reviewer 3 Report
The authors submitted a very interesting paper. Delirium is a form of acute brain dysfunction associated with increased morbidity and mortality. As the authors emphasized, acute systemic inflammation is a key driver of delirium in cases of acute illnesses. In particular, this narrative review focused on the altered kynurenine pathway (KP), which is a molecular pathway implicated in the pathogenesis of delirium. Importantly, the KP is highly regulated in the immune system and influences neurological functions. Since the activation of specific KP neuroactive metabolites could play a role in the event of delirium, the authors collectively describe the roles of the KP and speculate its relevance in delirium.
Overall, the manuscript provides a concise and informative overview of the importance of delirium and the potential role of the altered KP in its pathogenesis. I only have a few suggestions for improving the text.
1. Postoperative delirium is an important issue and neuroinflammation is a putative mechanism underlying its pathophysiology. It should be better discussed (doi: 10.23736/S0375-9393.17.12146-2.)
2. I suggest to divide research perspectives and conclusions in separate paragraph. Moreover, the research perspectives should be summarized into one or more tables. What about translational perspectives? In this section (i.e., perspectives) the authors should better underline that several biomarkers have been investigated for their potential utility in delirium, including inflammatory markers, neuroimaging measures, and markers of neuronal injury. In fact, inflammatory markers, such as cytokines and acute-phase reactants, have been found to be elevated in patients with delirium, particularly those with delirium associated with systemic inflammation, such as sepsis ( doi: 10.23736/S0375-9393.17.12146-2.). It should be better addressed.
3. Neuroimaging studies have identified structural and functional changes in the brain associated with delirium, including alterations in connectivity and metabolism. Finally, markers of neuronal injury, such as S100B (doi: 10.1016/j.bja.2022.01.005.) and neuron-specific enolase, have been found to be elevated in patients with delirium, particularly those with delirium associated with traumatic brain injury or in postoperative delirium (doi: 10.1371/journal.pone.0259217.).
4. Please, include potential limitations in the clinical use of these preclinical findings. In fact, while these biomarkers show promise in identifying delirium, there are limitations to their use. For example, the elevated levels of inflammatory markers may not be specific to delirium and may be seen in other conditions associated with inflammation (doi: 10.1056/NEJMoa1301372). Additionally, some biomarkers may require specialized equipment or invasive procedures, which limit their utility in clinical settings.
5. About clinical use. Please consider that previous studies do not found lower tryptophan levels in delirium on which the tryptophan depletion theory is based. However, it seems that a preoperative higher kynurenine/tryptophan ratio could be indicative of delirium (doi: 10.1016/j.psym.2011.09.009.).
6. Can you summarize differences between neuroinflammation and systemic inflammation in a table?
7.include legends and abbreviations in each table.
8. Include the suggested references.
Round 2
Reviewer 1 Report
The authors modify the manuscript per comments. The current version has met the standards of the journal ijms. And I recommentd accept in present form.